# Natural History of the Genus *Elasmoderus* Saussure, 1888 (Orthoptera: Tristiridae), an Endemic and Eremic Element of the Transitional Coastal Desert of Chile

**DOI:** 10.3390/insects15070513

**Published:** 2024-07-09

**Authors:** Mario Pizarro-Luna, Fermín M. Alfaro, Mario Elgueta, Jaime Pizarro-Araya

**Affiliations:** 1Carrera de Ingeniería Agronómica, Departamento de Agronomía, Escuela de Agronomía, Facultad de Ciencias, Universidad de La Serena, Casilla 554, La Serena, Chile; mpizarrol@alumnosuls.cl; 2Laboratorio de Entomología Ecológica (LEULS), Departamento de Biología, Facultad de Ciencias, Universidad de La Serena, Casilla 554, La Serena, Chile; fmalfaro@userena.cl; 3Doctorado en Biología y Ecología Aplicada, Universidad Católica del Norte, Universidad de La Serena, Casilla 554, La Serena, Chile; 4Área de Entomología, Museo Nacional de Historia Natural, Casilla 787, Santiago, Chile; mario.elgueta@mnhn.gob.cl; 5Doctorado en Conservación y Gestión de la Biodiversidad, Facultad de Ciencias, Universidad Santo Tomás, Ejército 146, Santiago, Chile; 6Instituto de Ecología y Biodiversidad (IEB), Ñuñoa, Santiago, Chile

**Keywords:** Atacama, coastal desert, coastal lomas, grasshoppers, Tristirinae, population outbreaks

## Abstract

**Simple Summary:**

The genus *Elasmoderus* is a group of the orthopteran family Tristiridae distributed across the transitional region of South America, Andean areas of Peru, Argentina, and Chile, the low-altitude desert and semidesert territories of Chile, and the steppe areas of the Argentinian and Chilean Patagonia. This genus is endemic to the extreme arid environments of north-central Chile and is famous for its adaptations to such habitats. Based on original and published data, we characterize the geographical distribution and discuss the history and ecology of this genus. Within this genus, *Elasmoderus lutescens* is the most widely distributed species, mainly in coastal and interior environments, followed by *E. minutus*, a small-sized species with a more restricted distribution range that is currently categorized as Vulnerable in Chile. Another relevant species is *Elasmoderus wagenknechti*, which is concentrated in coastal and interior environments of the Coquimbo region, where it can be very abundant. This species is economically significant and can damage crops and natural vegetation. Our findings highlight the importance of researching native insects to understand the role they play in semiarid ecosystems and to develop a basis for conducting long-term studies in northern Chile.

**Abstract:**

The genus *Elasmoderus* belongs to the family Tristiridae, an orthopteran group that is distributed across the transitional region of South America and represented in the Andean areas of Peru, Argentina, and Chile. The species of *Elasmoderus* have morphological adaptations that make them especially suited for surviving in extreme arid environments and are an endemic and eremic group of the north-central region of Chile. On the basis of field samplings, direct observations, and a comprehensive literature review, we collected information about the genus *Elasmoderus*. The objectives of this research were: (i) to provide updated information on the geographical distribution range of the species of the genus *Elasmoderus*, and (ii) to collect and synthesize the most relevant information on the natural history and ecology of this orthopteran group as a basis for future long-term studies of its populations. Although this genus is underrepresented compared to other orthopteran families, it has developed interesting adaptations to extreme arid environments that remain unstudied to this date. *E. lutescens* is known to be one of the species with a wider latitudinal distribution, mostly on the coast and longitudinal valley of Atacama. *E. minutus* has a more restricted distribution, whereas *E. wagenknechti* is concentrated in coastal and interior environments of the Coquimbo region, an area where it reaches high population densities, thus affecting crops and natural vegetation and polluting water sources. Our findings underscore the importance of conducting comprehensive research on native insect groups that are poorly known but crucial for arid and semiarid ecosystems. These data will serve as a starting point for conducting long-term studies on this orthopteran group to gain a better understanding of the importance and role of these species in the semiarid ecosystems of northern Chile.

## 1. Introduction

In Chile, the order Orthoptera is represented by 149 species grouped into 69 genera and 13 families [1,2,3] that are distributed along the entire Chilean continental territory from sea level up to above 4000 m. [1]. Approximately 56% of these genera and 75% of the species are endemic [2]. Most of the endemic species are concentrated in the Chilean desert, as is the case of the family Tristiridae, an eremic group that is characteristic of arid zones [4,5,6,7].

The family Tristiridae is a monophyletic group in the superfamily Acridoidea and is endemic to South America. It consists of 25 species grouped into the subfamilies Atacamacridinae, with a monotypic genus, and Tristirinae, with 3 tribes and 17 genera [5,6,7]. Its range encompasses part of the Andean region [8] as well as the Subantarctic subregion, central Chile, Patagonia, and the South American transitional area in Perú, Argentina, and Chile along the Andes mountain range from Tierra del Fuego to 10° S and from the Atlantic coast of Patagonia in the east to the Pacific coast in the west [4,5,6,7,9,10]. Most of the species in this family inhabit extremely arid environments, and the Chilean representatives are of medium to small size and have dull, brownish-grayish coloration, with rugged, tuberculated, and carinated tegument [5], and consist of 11 genera [10]. The range of the genus encompasses the Norte Chico region of Chile, from 28° to 32° S [7,10,11], i.e., approximately from Taltal (Antofagasta region) to Illapel (Coquimbo region) (Figure 1).

One of the most interesting aspects of the ecology of these groups has to do with the adaptations of their populations to extreme environmental conditions, such as those found in deserts [12,13]. According to Donato [4], the species of *Elasmoderus,* as well as the monospecific genera *Uretacris* Liebermann and *Enodisomacris* Cigliano, originated in the desert areas of South America as a result of vicariant events that split the area of Patagonia toward the end of the Miocene. This shows how current diversity patterns in this group may have been shaped by historical geological and climate events and helps us understand how these species have developed notable adaptations to unique extreme environments in response [14].

The genus *Elasmoderus* has developed interesting adaptations to extreme arid environments that remain unstudied to date. Also, one of its species with meridional distribution is one of the few endemic species in Chile that can cause economic damage to crops in the semiarid area of the Norte Chico [15,16]. These data present valuable opportunities to study grasshoppers that are lesser known but of considerable importance in arid and semiarid ecosystems. The objectives of this research study were (i) to provide updated information on the geographical distribution range of the species of the genus *Elasmoderus* and (ii) to collect and synthesize the most relevant information on the natural history and ecology of this orthopteran group as a potential starting point for future long-term studies of its populations.

## 2. Materials and Methods

### 2.1. Spatial Records and Geographical Distribution of Elasmoderus

Using guided samplings, we studied populations of species of *Elasmoderus* in multiple locations of the Antofagasta, Atacama, and Coquimbo regions (Chile) between 2010 and 2022. To examine the natural history and ecology of the genus, we visited locations where population outbreaks of species such as *E. lutescens* and *E. wagenknechti* have occurred during rainy years (2015, 2022) and in periods coinciding with the flowering desert phenomenon.

The range of *Elasmoderus* was identified based on a review of reference material from the following collections: the entomological collection of Universidad de Tarapacá, Arica, Chile (IDEA, Héctor A. Vargas); the entomological collection of Laboratorio de Entomología Ecológica of the Departamento de Biología of Universidad de La Serena, La Serena, Chile (LEULS, Jaime Pizarro-Araya); Instituto de Entomología, Universidad Metropolitana de Ciencias de la Educación, Santiago, Chile (IEU-MCE, Patricia Estrada); Museo Argentino de Ciencias Naturales Bernardino Rivadavia, Buenos Aires, Argentina (MACN-Ar, Pablo Mullieri); División Entomología, Museo de La Plata, La Plata, Argentina (MLP, Pablo M. Dellapé); National Museum of Natural History, Santiago, Chile (MNHN, Mario Elgueta); Zoological Museum of Universidad de Concepción, Concepción, Chile (MZUC, Juan Carlos Ortiz).

The collection records were supplemented with both original materials obtained by pitfall traps and manual captures conducted by some of the authors between 2004 and 2022 and the published data (i.e., 1,10,11,15,16) (Appendix A). The collected material was identified at the species level using keys and descriptions by Cigliano [9] and Cigliano et al. [10]. All collected specimens were deposited in the entomological collection of the Laboratorio de Entomología Ecológica of Universidad de La Serena, Chile (LEULS) and the Entomology section of Chile’s National Museum of Natural History, Santiago, Chile (MNNC, Mario Elgueta).

### 2.2. Distribution Mapping

All the sampled localities were georeferenced using a GPS receiver (Etrex-Personal Navigator, Garmin). For each individual collected in the field, its geographical coordinate was recorded with the GPS equipment after being turned on for at least 5 min. For the collected individuals, the geographic coordinate was estimated from the Google Earth Pro search. The distribution maps were generated using ArcGIS 9.3 based on the records generated during the study and from the collected databases. The maps were built from satellite images of the study area (1:250,000 scale) and used the regional limits included in that area. The satellite image corresponds to Landsat 7, introducing the Enhanced Thematic Mapper Plus and a new panchromatic band to continue the legacy of Earth Observation. The image was downloaded from https://earthexplorer.usgs.gov/ (accessed on 27 March 2024). The regional limits and centroids of the capitals correspond to shape files provided free of charge since the use we give them is educational and research by the Military Geographic Institute of Chile [17]. The layers are at a resolution of 1:250,000. All the maps were georeferenced using UTM Datum WGS 84/zone 19S (EPSG: 32179, EPSG).

## 3. Results

### 3.1. Distribution Patterns and Habitat Preference in Elasmoderus

The genus *Elasmoderus* is an endemic and eremic group of northern Chile that is restricted to the strip extending from 26° to 32° S, a region that encompasses part of the biogeographical provinces of Atacama and Coquimbo [18]. According to Morrone [8,17], the biogeographical province of Atacama, located between 18° and 28° S and embedded in the transitional region of South America, is inhabited by the species of *Elasmoderus* as some of its characteristic elements.

The biogeographical province of Coquimbo is part of the Central Chile subregion of the Andean region and encompasses the north-central region of Chile from 28° to 32° S [18], where *Elasmoderus lutescens* and *E. wagenknechti* are found.

Now, *Elasmoderus* is distributed over the coastal environments from Taltal (Antofagasta region) to Illapel (Coquimbo region), a longitudinal strip that extends approximately from 26° to 32° S. This territory includes protected areas such as the Pan de Azúcar National Park, Llanos de Challe National Park, and Fray Jorge National Park (see Appendix A). Altitudinally, the genus extends from virtually sea level to approximately 3089 m in the mountain range of the Atacama region (Figure 1). Regarding *E. lutescens* (Figure 2A and Figure 3A,B), based on a total of 59 records, it was found that this species inhabits a variety of plain and mountain shrub habitats of the Desierto Florido in Atacama. We also collected isolated specimens in the semiarid basins of Atacama and Coquimbo, more specifically in the valleys of Huasco, Elqui, and Limarí. These collection campaigns were conducted at altitudes ranging from sea level up to 1100 m and reflect a larger range for this genus. *E. minutus* (Figure 2B and Figure 3C,D) is apparently a scarcer species, with only 6 known records, and inhabits the Pan de Azúcar National Park to the interior coastal areas of the Atacama desert. As for *Elasmoderus wagenknechti* (Figure 2C and Figure 3E,F), we gathered 46 records, all of them from different locations of the Choapa and Limarí provinces (Coquimbo region), and its distribution is associated with the interior dryland of these provinces. The records show distribution at altitudes ranging from 450 m to 1300 m. The only specimen was found in Illapel, Choapa province, as well.

### 3.2. Natural History and Ecology of Elasmoderus

The study of the genus *Elasmoderus* is of particular interest as it includes both the species with a wide latitudinal distribution in northern Chile and the species with a more limited distribution, which, for that exact reason, is currently classified as Vulnerable [19]. It also contains the third species that is considered a potential crop pest for interior dryland areas of the Coquimbo region. The species of *Elasmoderus* inhabit mainly arid and semiarid environments of coastal and low-lying areas (1000–1200 m.a.s.l.) in areas dominated by plant formations consisting of herbaceous plants (annuals and perennials) and shrubs; records from high mountainous areas are scarce (Figure 2A,C).

The adults of *E. lutescens* (Figure 3A,B) are macropterous, with the head as wide as the pronotum and the posterior wings semitransparent with black veins; the males are approximately 20 mm in length, and the females are 30 mm in length [10]. In the margins of the Atacama region, during seasons with above-average annual precipitation (ENSO years, El Niño Southern Oscillation), significant population outbreaks of this species are commonly observed, especially in semiarid areas of the southern part of the region (Figure 4A,C). During some of these events, oviposition has been observed in soils with low vegetation cover and times of the day when temperatures reach above 70 °C (Figure 4A). The adults are an abundant but temporal food resource for other insects, such as native predator carabids (Figure 4C).

*Elasmoderus minutus* is the smallest species in the genus and is also macropterous. Adult males are 12–14 mm in length, and females are 20–24 mm in length. This species prefers habitats located in hillsides, plains, and pasture areas of the semiarid Atacama region [10]; no crop damage has been reported. According to Cigliano et al. [10], the species is distributed across the periarid Mediterranean biogeographical region of Atacama, more specifically in the area extending from 26° to 28° S and at altitudes between 22 m and 390 m. In 2014, the members of a Chilean–Argentinean team (Laboratorio de Entomología Ecológica, Universidad de La Serena (LEULS)–Museo Argentino de Ciencias Naturales Bernardino Rivadavia (MACN)) found a male specimen during an expedition to the Atacama mountain range. The specimen was found in a semiarid environment located on the road to Pascua Lama at an altitude of 3280 m. (see Appendix A). This is the highest known record for this species and genus (unpublished data) and extends the distribution range of *E. minutus* towards the central-southern margin of the Atacama region.

*Elasmoderus minutus* is currently classified as Vulnerable in the Chilean list of endangered species [20]. This status is based on several parameters, such as having a distribution range under 20,000 km^2^ and being present in fewer than 11 locations. Its population has declined as a result of the destruction of its habitat caused by human activities such as the extraction of native wood and the loss of vegetation cover, construction (road building and improvement), mining activities, and low precipitation [21]. Additionally, in recent years, low rainfall has impacted its habitat, and mining activities are an imminent threat that can damage the environment and affect the vegetation cover, which sustains the large diversity of species in these semiarid areas [22]. It is worth noting that there are other latent threats, including road construction, such as the road connecting Taltal and Chañaral, which have negatively impacted the environments occupied by the populations of the species. There are climate changes directly affecting this species and its environment, as well [23].

In *Elasmoderus wagenknechti* (Figure 3E,F), adults are larger than in other species of the genus and are characterized by short forewings and absent hind ones. Male adults are 23–27 mm in length, and female adults measure 33–37 mm [10]. This species inhabits the interior areas of the south-central portion of the Coquimbo region; however, young adults have been recorded [1] in the Antofagasta (Taltal) and Atacama regions (Carrizal Bajo, Hacienda Castilla, Caserones, Fundo Marañón) as well, but their identification should be confirmed. The area where this species has a confirmed presence is characterized by flat lands with low slope gradients that are connected to low ridges interspersed by small ravines and narrow valleys. The soils are aridisols with coarse texture and poorly developed. The climate is the subtropical semiarid Mediterranean, with hot and dry summers and mild and humid winters. Annual precipitation is about 210 mm, though it varies widely between years, with humid years associated with the ENSO phenomenon. *E. wagenknechti* is locally known as ‘langosta de Combarbalá’ (Combarbalá locust) because of the sudden population outbreaks that have taken place in areas near the locality of the same name. However, these outbreaks have not been widespread in all areas near the city since some places are more favorable for this species than others (Figure 5A,C). According to local reports, these population outbreaks are recurrent in the north-central area of Chile, although there are no historical data available, except for the records by Moroni [24] and Toro [25], who reported densities of 10–50 spec/m^2^, and Cepeda-Pizarro et al. [15,16], who reported densities of 0.25–0.54 spec/m^2^. No high densities of *E. wagenknechti* have been reported during recent years, which may be a result of the low rainfall received in the area and region during dry years. It has been reported that in 2020, an outbreak occurred in the locality of Agua Amarilla with an estimated density of 10–20 spec/m^2^ in September; more recently, in 2022, densities ranging from 15 to 30 spec/m^2^ were measured in the area of Divisadero, Limarí (30°50′33″ S, 71°09′19″ W, 1100 m.a.s.l.).

In general, the species of *Elasmoderus* can be an important element within the trophic nets of the desert ecosystems. On the one hand, they are food for vertebrates, and on the other, they are primary consumers and destroyers of organic matter. More specifically, *E. wagenknechti*, whose population dynamics are closely linked to specific climate factors, can negatively affect both natural and cultivated plants. Some of these negative effects include (i) damage to crops (especially vegetables) because during these outbreaks, the individuals feed mainly on plant shoots, which may cause significant damage to subsistence farming in the interior dryland of the Limarí province; (ii) competition with native species, as these outbreaks may cause competition with other native orthopterans sharing the same habitat, such as species from various genera of Tettigoniidae (e.g., *Anysophia* Karabag, 1960, *Barraza* Koçak and Kemal, 2008, *Platydecticus* Chopard, 1951, *Polycleptis* Karsch, 1891); (iii) impact on plant diversity, since these outbreaks may affect the region’s biodiversity by preferring certain types of native plants, thus reducing food and habitat availability for other species, which affects the richness of local flora and fauna and reduces seed availability for many native species.

Additionally, the discussed species may be excellent models for population monitoring and life table studies. These approaches will help better understand how climate variability at the local scale may impact the population dynamics of native species and, consequently, economic activities, such as small-scale farming.

## 4. Conclusions

Studying the geographical distribution of the species of *Elasmoderus* in northern Chile is essential for understanding their occurrence in different local habitats in the transitional coastal desert of Chile. In addition to their importance for understanding the species distribution patterns in desert environments, these data can play a significant role in future studies of the ecology and conservation biology of the model species. It will also be essential for making informed decisions about the preservation of their fragile habitats and understanding their significance as pests in the interior drylands of the Coquimbo region. These aspects are critical for the conservation of biodiversity, as they provide an understanding of how species can adapt and survive in extreme conditions and have significant implications for conservation efforts associated with similar ecosystems all over the world.

The classification of *E. minutus* as Vulnerable implies that the species is at risk of extinction in the near future, the main conservation action being the management of the habitat, which is currently affected by anthropogenic activity and resource extraction. Also, since *E. wagenknechti* can reach high population densities and affect crops and native vegetation, it is essential to understand what climate and environmental factors may lead a native species to become one that causes economic damage to dryland crops. Hence, the synthesis of information about the natural history and ecology of *Elasmoderus* provides a basis for future population and behavioral studies. This is particularly relevant in the context of climate change, where a dramatic decrease in precipitation has been observed in the past 20 years, and habitat loss is impacting arid and semiarid ecosystems. A deeper understanding of these native insects and their ecological role may help us better understand how these ecosystems work and would provide essential information for developing more effective management and conservation strategies.

The genus *Elasmoderus*, as an endemic and eremic element of the transitional coastal desert of Chile, plays a crucial role in the structure and function of these local ecosystems. An up-to-date understanding of its geographical distribution, natural history, and ecology provides a solid starting point for future long-term studies aimed at the sustainable management and conservation of these important biological resources. We emphasize the need to implement effective conservation strategies that ensure the preservation of its natural habitat and the protection of its unique biodiversity in the context of climate change and growing anthropogenic pressures in the region.

## Figures and Tables

**Figure 1 insects-15-00513-f001:**
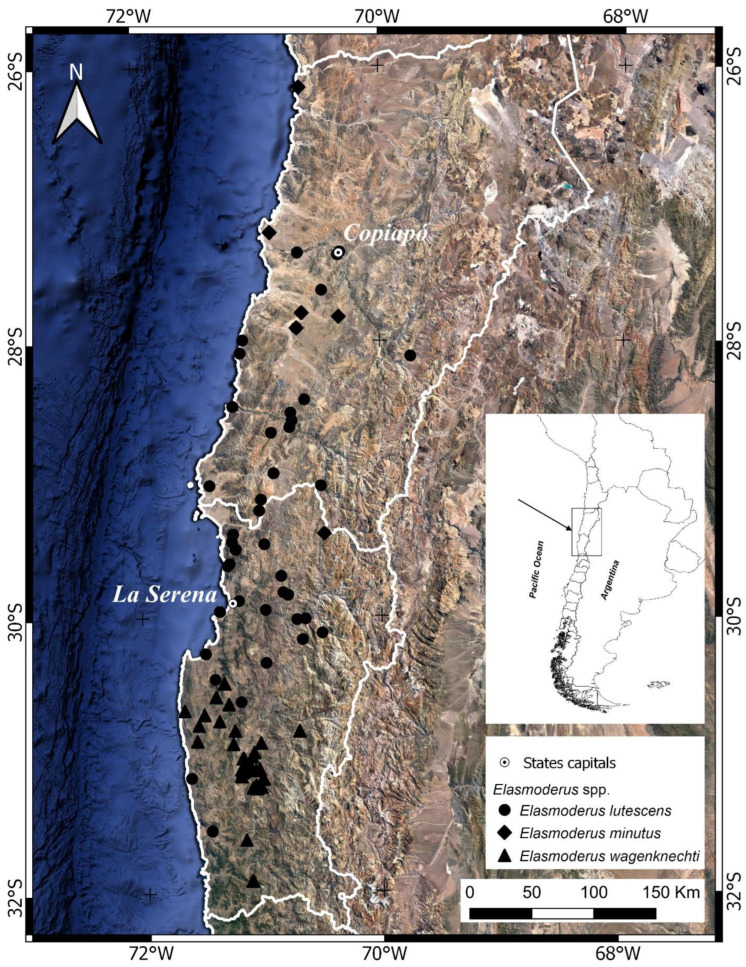
Distribution map of spatial records of the genus *Elasmoderus* Saussure, 1888 (Orthoptera: Tristiridae) and its corresponding species.

**Figure 2 insects-15-00513-f002:**
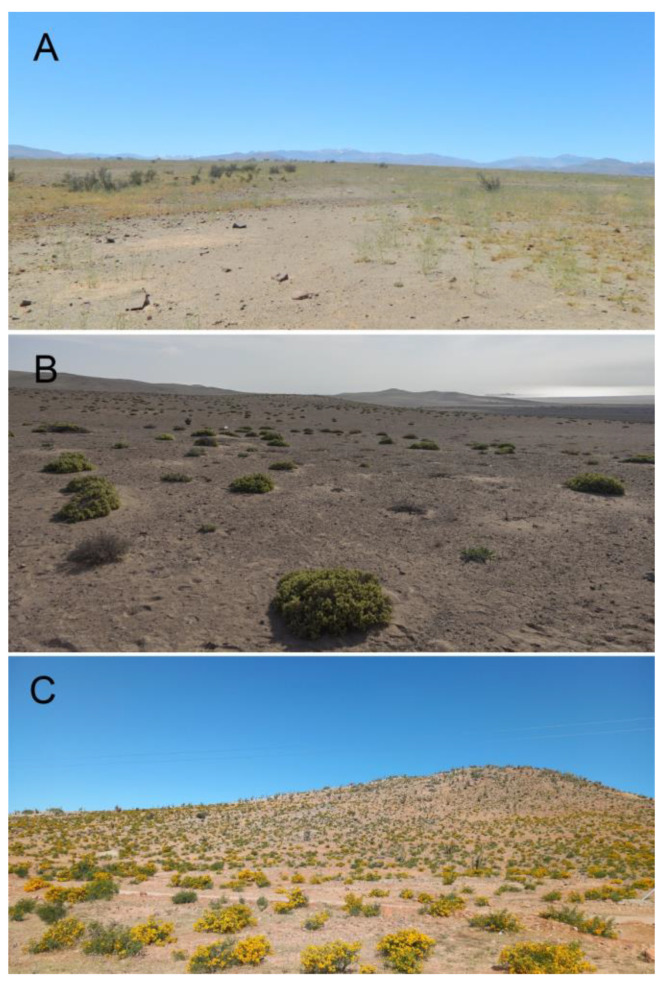
Habitats with species of *Elasmoderus* in Chile. (**A**) Habitat of *Elasmoderus lutescens* in Llanos de Pajonales, in the south of the Atacama region; (**B**) Habitat of *Elasmoderus minutus* in flatlands of the former Hacienda Castilla, in the north of the Atacama region; and (**C**) Habitat of *Elasmoderus wagenknechti* in Divisadero (Combarbalá, Coquimbo region).

**Figure 3 insects-15-00513-f003:**
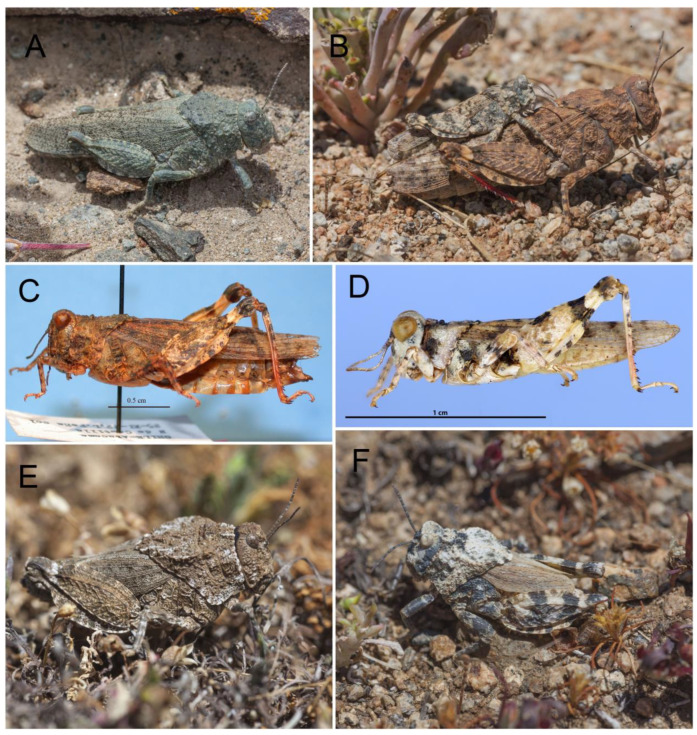
Individuals of *Elasmoderus*. (**A**) Side view of *Elasmoderus lutescens* in Boquerón Chañar (Atacama region); (**B**) side view of two individuals of *Elasmoderus lutescens* copulating (the smaller-sized male is on top of the female); (**C**) side view of the female (Allotype) of *Elasmoderus minutus* deposited in the collection of Museo de La Plata, Facultad de Ciencias Naturales y Museo (La Plata, Argentina) (taken from Cigliano et al. [7]); (**D**) side view of the male (Holotype) of *Elasmoderus minutus* deposited in the collection of Museo de La Plata, Facultad de Ciencias Naturales y Museo (La Plata, Argentina) (taken from Cigliano et al. [7]; (**E**) side view of a female of *Elasmoderus wagenknechti* in Peñablanca (Coquimbo region); (**F**) side view of a male of *Elasmoderus wagenknechti* in Peñablanca (Coquimbo region).

**Figure 4 insects-15-00513-f004:**
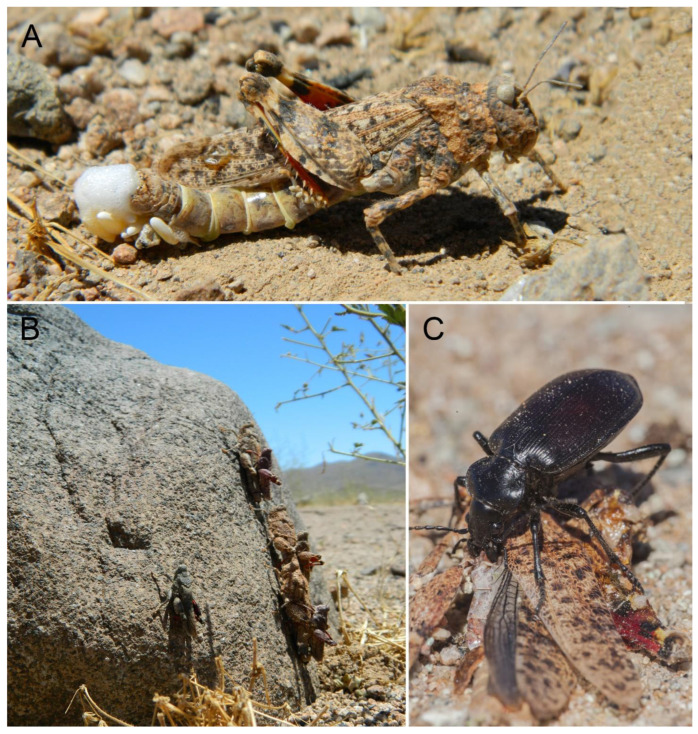
Different aspects of the natural history of *Elasmoderus*. (**A**) Side view of a female *Elasmoderus lutescens* laying eggs in soils of Llanos de Pajonales (Atacama region); (**B**) Individuals of *Elasmoderus lutescens* copulating in Llanos de Pajonales (Atacama region); (**C**) Individual of *Calosoma vagans* (Coleoptera: Carabidae) feeding on a female *Elasmoderus lutescens* in Llanos de Pajonales (Atacama region).

**Figure 5 insects-15-00513-f005:**
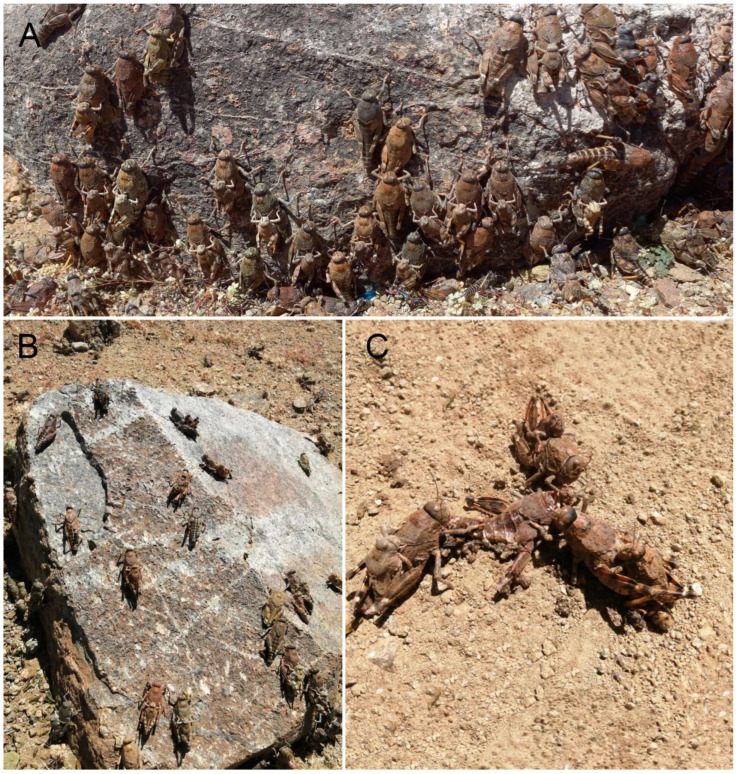
Different aspects of the natural history of *Elasmoderus*. (**A**) Group of individuals of *Elasmoderus wagenknechti* during the outbreak in Divisadero (Combarbalá, Coquimbo region); (**B**) individuals of *Elasmoderus wagenknechti* copulating; (**C**) individuals of *Elasmoderus wagenknechti* copulating while feeding on the dead specimen.

## Data Availability

The collected specimens have been deposited in the entomological collection of Laboratorio de Entomología Ecológica of Universidad de La Serena (LEULS). All published data are available upon formal request.

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
