# Peer review of "Natural History of the Genus Elasmoderus Saussure, 1888 (Orthoptera: Tristiridae), an Endemic and Eremic Element of the Transitional Coastal Desert of Chile"

_insects, 2024, doi:10.3390/insects15070513_

Round 1
Reviewer 1 Report (Previous Reviewer 2)
Comments and Suggestions for Authors
OK
Author Response
Dear Dr Ivana Vostic
Assigned Editor:
We appreciate your valuable suggestions and corrections of our article titled Natural History of the Genus Elasmoderus Saussure, 1888 (Orthoptera: Tristiridae), an Endemic and Eremic Element of the Transition Coastal Desert of Chile to the special issue Locusts and Grasshoppers: Bionomics, Distribution, and Population Management. In this new version (word document with change control) we have largely incorporated all the changes and suggestions made by you and the reviewers. Below you will find the responses to each of the comments and suggestions from the reviewers and the editor. All remaining changes proposed by the reviewer have been included directly in the document and have been marked with tracked changes in the first revised version of this manuscript.
In biogeography, usually "transition"
Response. Done
Please, use em-dashes where it is necessary
Response. Done
you can remove "distribution" in almost all cases
Response. Done
I am sure that this subsection should be moved to the Section 2
Response. Done. We have moved this section to the beginning of Materials and Methods (lines 114-120).
you may remove a.s.l. in almost all cases
Response. Done
– I suggest to re-write this part.
(1) To add the special Subsection (e.g. 2.2. Distribution mapping).
Response. Done (see line 142)
(2) To describe how you determine coordinates of all specimens (especially from the collections).
Response. All the sampled localities were georeferenced using a GPS receiver (Etrex-Personal Navigator, Garmin). For each individual collected in the field, its geographical coordinate was recorded with the GPS equipment after being turned on for at least 5 minutes. For the collection individuals, the geographic coordinate was estimated from the Google Earth Pro search.
(3) If you used some sattelite images, please, describe their parameters (source, format, resolution).
Response. The satellite image corresponds to Landsat 7, introducing the Enhanced Thematic Mapper Plus, and a new panchromatic band to continue the legacy of Earth Observation. The image was downloaded from https://earthexplorer.usgs.gov/. The regional limits and centroids of the capitals correspond to shape files provided free of charge, since the use we give them is educational and research, by the Military Geographic Institute of Chile (IGM, https://www.igm.cl /). These layers are at a resolution of 1:250,000.
(4) You write about the scale 1:250,000... for what? the map in the text is 1:5,000,000.
Response. The layers are at a resolution of 1:250,000, but the map is displayed at a scale of 1:5,000,000.
(5) You write that you used UTM Datum WGS 84 / zone 19S. - For what?
Response. Because for the Chilean zone, UTM Datum WGS 84 / zone 19S is used, which corresponds to continental Chile (see https://www.geofumadas.com/las-coordenadas-utm-en-el-hemisferio-sur/#:~:text=Argentina%3A%2018%2C19.20%2C21,Uruguay%3A%2021%2C22).
(6) If you produced the basic map, please, describe it in very explicit manner.
Response. We have tried to be as explicit as possible in the description of the map's production. In general terms, a shape file was created from the spatial records database, which was then compiled with the satellite image and the limits of the study area, all of this in ArcGis. See below (see lines 147-150):
The distribution maps were generated using ArcGIS 9.3 based on the records generated during the study and from the collected databases. The maps were built from satellite images of the study area (1:250,000 scale) and used the regional limits included in that area.
(7) And, as a result, to characterize the basic map (e.g. A Universal Transverse Mercator projection and World Geodetic System 84 ellipsoid (ETRS ##) was used to produce the basic map.)
Response. The ETRS code has no equivalence outside Europe, but in the case of UTM 19S WGS 84, the international code used by us is the following: EPSG:32179, EPSG = European Petroleum Survey Group and corresponds to an SRID (Spatial Reference System Identifier) ​​or Spatial Reference Identifier.
Please, clarify... For instance, I understand this phrase as "plain and mountain shrub habitats of the Desierto Frolido...", but I am not sure that this understanding is correct
Response. Done. We have clarified the phrase according to what the reviewer indicates (see lines 201-203).
please, clarify, because herbaceous plants includes annual plants...
Response. Done. We have included annuals and perennials within herbaceous plants.
You write about egg predation, but use to illustrate it the adult ground beetle eating the adult grasshopper!
Response. Thank you for your observation, we have corrected that section.
(1) Please, re-write this part. (2) May this occurrence be associated with human activities? I mean this specimens may travel with some vehicle from low altitudes.
Response. This is the only record that we have in our laboratory. Our knowledge of the group in general has shown us that this species in particular is very rare to observe in collections. We think it is unlikely that individuals of this species could be transported by human actions.
Please, clarify. Do you mean yound adults or hoppers?
Response. Yound adults.
or - its occurence should be confirmed
Response. Done.
Please, add authors
Response. Done.
Response. Done. We have clarified the phrase according to what the reviewer indicates (see lines 355-357).
The classification of E. minutus as Vulnerable implies that the species is on the risk of extinction in the near future, the main conservation action being the management of the habitat, which is currently affected by anthropogenic activity and resource extraction.
May you discuss what conservation actions are prospective for the species?
Response. Done. We have clarified the phrase according to what the reviewer indicates (see lines 355-357).

Reviewer 2 Report (Previous Reviewer 3)
Comments and Suggestions for Authors
The authors have improved the manuscript by adding a table of all localities, which also states the newly collected data. This would, in my view, make this worth publishing as a natural history short note that could fit on two pages in a data journal. I.e., the format of this article is wrong.
I feel like I must apologize to the authors because this may be an issue of incongruence between my view as a referee and the journal policy. So, I leave it up to the discretion of the editor.
Kind regards
Author Response
Dear Dr Ivana Vostic
Assigned Editor:
We appreciate your valuable suggestions and corrections of our article titled Natural History of the Genus Elasmoderus Saussure, 1888 (Orthoptera: Tristiridae), an Endemic and Eremic Element of the Transition Coastal Desert of Chile to the special issue Locusts and Grasshoppers: Bionomics, Distribution, and Population Management. In this new version (word document with change control) we have largely incorporated all the changes and suggestions made by you and the reviewers. Below you will find the responses to each of the comments and suggestions from the reviewers and the editor. All remaining changes proposed by the reviewer have been included directly in the document and have been marked with tracked changes in the first revised version of this manuscript.
In biogeography, usually "transition"
Response. Done
Please, use em-dashes where it is necessary
Response. Done
you can remove "distribution" in almost all cases
Response. Done
I am sure that this subsection should be moved to the Section 2
Response. Done. We have moved this section to the beginning of Materials and Methods (lines 114-120).
you may remove a.s.l. in almost all cases
Response. Done
– I suggest to re-write this part.
(1) To add the special Subsection (e.g. 2.2. Distribution mapping).
Response. Done (see line 142)
(2) To describe how you determine coordinates of all specimens (especially from the collections).
Response. All the sampled localities were georeferenced using a GPS receiver (Etrex-Personal Navigator, Garmin). For each individual collected in the field, its geographical coordinate was recorded with the GPS equipment after being turned on for at least 5 minutes. For the collection individuals, the geographic coordinate was estimated from the Google Earth Pro search.
(3) If you used some sattelite images, please, describe their parameters (source, format, resolution).
Response. The satellite image corresponds to Landsat 7, introducing the Enhanced Thematic Mapper Plus, and a new panchromatic band to continue the legacy of Earth Observation. The image was downloaded from https://earthexplorer.usgs.gov/. The regional limits and centroids of the capitals correspond to shape files provided free of charge, since the use we give them is educational and research, by the Military Geographic Institute of Chile (IGM, https://www.igm.cl /). These layers are at a resolution of 1:250,000.
(4) You write about the scale 1:250,000... for what? the map in the text is 1:5,000,000.
Response. The layers are at a resolution of 1:250,000, but the map is displayed at a scale of 1:5,000,000.
(5) You write that you used UTM Datum WGS 84 / zone 19S. - For what?
Response. Because for the Chilean zone, UTM Datum WGS 84 / zone 19S is used, which corresponds to continental Chile (see https://www.geofumadas.com/las-coordenadas-utm-en-el-hemisferio-sur/#:~:text=Argentina%3A%2018%2C19.20%2C21,Uruguay%3A%2021%2C22).
(6) If you produced the basic map, please, describe it in very explicit manner.
Response. We have tried to be as explicit as possible in the description of the map's production. In general terms, a shape file was created from the spatial records database, which was then compiled with the satellite image and the limits of the study area, all of this in ArcGis. See below (see lines 147-150):
The distribution maps were generated using ArcGIS 9.3 based on the records generated during the study and from the collected databases. The maps were built from satellite images of the study area (1:250,000 scale) and used the regional limits included in that area.
(7) And, as a result, to characterize the basic map (e.g. A Universal Transverse Mercator projection and World Geodetic System 84 ellipsoid (ETRS ##) was used to produce the basic map.)
Response. The ETRS code has no equivalence outside Europe, but in the case of UTM 19S WGS 84, the international code used by us is the following: EPSG:32179, EPSG = European Petroleum Survey Group and corresponds to an SRID (Spatial Reference System Identifier) ​​or Spatial Reference Identifier.
Please, clarify... For instance, I understand this phrase as "plain and mountain shrub habitats of the Desierto Frolido...", but I am not sure that this understanding is correct
Response. Done. We have clarified the phrase according to what the reviewer indicates (see lines 201-203).
please, clarify, because herbaceous plants includes annual plants...
Response. Done. We have included annuals and perennials within herbaceous plants.
You write about egg predation, but use to illustrate it the adult ground beetle eating the adult grasshopper!
Response. Thank you for your observation, we have corrected that section.
(1) Please, re-write this part. (2) May this occurrence be associated with human activities? I mean this specimens may travel with some vehicle from low altitudes.
Response. This is the only record that we have in our laboratory. Our knowledge of the group in general has shown us that this species in particular is very rare to observe in collections. We think it is unlikely that individuals of this species could be transported by human actions.
Please, clarify. Do you mean yound adults or hoppers?
Response. Yound adults.
or - its occurence should be confirmed
Response. Done.
Please, add authors
Response. Done.
Response. Done. We have clarified the phrase according to what the reviewer indicates (see lines 355-357).
The classification of E. minutus as Vulnerable implies that the species is on the risk of extinction in the near future, the main conservation action being the management of the habitat, which is currently affected by anthropogenic activity and resource extraction.
May you discuss what conservation actions are prospective for the species?
Response. Done. We have clarified the phrase according to what the reviewer indicates (see lines 355-357).

This manuscript is a resubmission of an earlier submission. The following is a list of the peer review reports and author responses from that submission.
Round 1
Reviewer 1 Report
Comments and Suggestions for Authors The methods section lacks detailed descriptions of the sampling and observation processes, and there is a lack of information on experimental design and data analysis. The authors should conduct a thorough analysis of the collected data from the investigation to elucidate the distribution patterns of the genus. This analysis should be presented in the form of charts and graphs to visually represent the findings. overall, this study conducted an investigation on the distribution characteristics of a particular genus in a specific region. The research methods and content are generally acceptable, but overall, it is a basic and ordinary research work. From the perspective of regional basic data, it holds significance, but it is limited to that scope and may not generate broader interest.Author Response
Please see the attachment.

Reviewer 2 Report
Comments and Suggestions for Authors
The authors present fine data on the distribution of the three species of genus Elasmoderus and give a review about their ecology with interesting photos of their habitat and biology. Everything is done quite nicely. There is only one more serious problem concerning the distribution of E. wagenknechti. According to OSF (cited by the authors) the species is found much more southern (see map in OSF; with specimen links!) than indicated in the paper. The origin and reason of this discrepancy must be clarified.
Some small problems
- the first part of the abstract is too long and not suited to the topic
- l 67 why not [5–7,3,4] -> [3–7]?. Please give the references in numerical order (problem at several places in the ms)
- grammar: Figure 1. Distribution map of spatial records of the genus Elasmoderus Saussure, 1888 (Orthoptera: Tristiridae) and _its_their corresponding species.
- abstract +l94: not a proper comparison - genus versus other orthopteran families
Reviewer 3 Report
Comments and Suggestions for Authors
The authors present a manuscript on the distribution and natural history of three species of Orthoptera in a region of Chile. I agree that distribution and natural history information in general merits publication due to its importance for conservation, management, and other fields. However, since the content of this manuscript seems to me like a topic of regional interest, I would suggest publication in a journal with a regional scope.
The writing of the manuscript is good. I think that the text is – as this is essentially a short note – somewhat lengthy, but this per se is not necessarily bad, as the authors filled the space with an ample literature review. However, the structure deserves, in my opinion, modification. It is not clear what the actual novelty of the manuscript is. The authors state that they collected distribution records from museums, but then they only present a map of all known distribution records – no coordinates, no metadata, which is the actually interesting bit. How many new records (per species / museum) were collected? The authors also state in the Data Availability statement that they collected some new specimens, but I cannot find this anywhere else in the text.
In short, the manuscript is in my opinion unnecessarily long, but at the same time missing the essential bits.
Since the authors do not state how much data they actually collected newly, it is difficult to judge to what extent this data merits publication. It may well be that it is worth publishing, but I find myself unable to judge this from what is given in the manuscript.
Kind regards
Oliver Hawlitschek
